# Exploring the Contribution of TLR7 to Sex-Based Disparities in Respiratory Syncytial Virus (RSV)-Induced Inflammation and Immunity

**DOI:** 10.3390/v17030428

**Published:** 2025-03-16

**Authors:** Mark A. Miles, Thomas D. Huttmann, Stella Liong, Felicia Liong, John J. O’Leary, Doug A. Brooks, Stavros Selemidis

**Affiliations:** 1Centre for Respiratory Science and Health, School of Health and Biomedical Sciences, RMIT University, Bundoora, VIC 3083, Australia; mark.miles@rmit.edu.au (M.A.M.); s3896003@student.rmit.edu.au (T.D.H.); stella.liong@rmit.edu.au (S.L.); felicia.liong@rmit.edu.au (F.L.); 2Discipline of Histopathology, School of Medicine, Trinity Translational Medicine Institute (TTMI), Trinity College Dublin, D08 XW7X Dublin, Ireland; solaoire@gmail.com; 3Sir Patrick Dun’s Laboratory, Central Pathology Laboratory, St James’s Hospital, D08 XW7X Dublin, Ireland; 4Clinical and Health Sciences, University of South Australia, Adelaide, SA 5001, Australia; doug.brooks@unisa.edu.au

**Keywords:** respiratory syncytial virus, toll-like receptor 7, inflammation, sex differences

## Abstract

TLR7 plays a key role in recognizing viral RNA to initiate an immune response. Sex-based differences in the severity of RSV respiratory infections have been noted, and this may be related to higher expression of X-linked toll-like receptor 7 (TLR7) in female immune cells. Indeed, TLR7 has been shown to influence sex differences in responses to other respiratory viruses; however, its role in RSV infection remains underexplored. We infected adult C57Bl/6 or TLR7 knockout mice with RSV and compared the specific lung immune responses between different sexes. Gene expression analysis revealed that infected female mice had elevated levels of type I and II interferons, proinflammatory cytokines, chemokines, and viral transcripts in their lungs compared to males. Additionally, females exhibited increased numbers of macrophages and higher antibody responses in the airways. Deletion of TLR7 diminished the sex differences in certain cytokine and antibody responses. Furthermore, ex vivo infection of male alveolar macrophages with RSV resulted in greater production of proinflammatory cytokines and viral transcripts than in female macrophages, suggesting inherent sex differences in macrophage responses. These findings provide new insights into the mechanisms underlying sex differences in RSV pathophysiology and suggest that TLR7 contributes to an enhanced inflammatory response in females.

## 1. Introduction

Respiratory syncytial virus (RSV) is a highly infectious negative-sense, single-stranded RNA (ssRNA) virus that infects millions of people globally. Although generally mild, RSV infection in vulnerable populations has the potential to cause severe respiratory illnesses such as bronchiolitis, pneumonia, and even death [1]. RSV infection is the leading cause of lower respiratory tract (LRT) disease in young children, accounting for 22% of all episodes of acute LRT infection in children globally [2,3]. Older adults, particularly those over 65 years of age or with comorbidities, are also susceptible to harmful outcomes from RSV infection, with 10% of severe respiratory infections in older adults due to RSV [4]. Despite the virus circulating in the global population for many decades, effective therapeutic intervention or prevention strategies remain limited and, until recently, have been primarily addressed to older age groups. The first RSV vaccines, Abrysvo, Arexvy, and mResvia, were recently approved for use in older adults, showing a significant reduction in RSV-associated LRT disease in over-60-year-olds [5,6,7]. For infants, vaccination during pregnancy offers some protection for newborns during their early months of life [8]. The prophylactic use of RSV-neutralizing monoclonal antibodies like Palivizumab and Nirsevimab can also provide protection in young infants, but these treatments are reserved for high-risk cases, and their effects diminish relatively quickly [9,10]. While these advancements highlight the progress in understanding RSV’s impact on vulnerable populations, it is important to explore additional factors, such as biological sex, which may influence disease severity, and the efficacy of the immune response mounted to RSV.

Sex is an important biological factor that can influence how RSV infection affects disease progression in an individual. Typically, respiratory viruses lead to more severe disease in males, although this pattern can vary depending on age [11]. This is probably because females generally can mount a more robust immune response than males [12], and this could involve X-linked gene expression. A sex bias for RSV infection has been observed mainly in younger age groups, impacting both acute and chronic illnesses related to the infection. For example, in children under 6 months of age, males are more likely to be hospitalized with RSV-associated acute respiratory distress [13,14,15,16]. Severe RSV infection is also a strong risk factor for chronic asthma and recurrent wheeze [17]. Asthma is almost twice as prevalent in boys than in girls and may reflect the sex-dependent altered immune lung environment due to early-life RSV infection [18]. Indeed, boys represented the higher proportion of recurrently wheezing children with RSV infection and presented higher blood eosinophil counts than girls [19]. Such findings have been complemented by a neonatal mouse model where early-life RSV infection resulted in severe allergic exacerbation later in life in males but not in females [20].

In adult and elderly cohorts, the incidence of RSV is slightly higher in females, although the hospitalization rates appear to be equally distributed between the sexes [21,22,23]. This argues against a sex bias in older individuals, at least for the severe manifestation of the disease, and may instead reflect other factors such as the increased contact that female caregivers have with young children, which can potentially facilitate the spread of the infection due to the higher prevalence of RSV in the young [24]. Differences in sex steroid hormones are also likely to contribute, although the disparity in hormone concentrations between males and females is less pronounced in adulthood compared to early stages of life, when significant sex differences in relation to RSV infection are observed [25]. Additionally, the natural aging immune system in older adults differs between sexes, with males displaying greater innate and proinflammatory activity but lower humoral responses [26]. This suggests that the immune responses to infection in older males may not be as efficient as in females, although their effect specifically in RSV infection remains underexplored. In addition, little mechanistic insight has been reported on how sex can influence RSV infection in adults.

Various genetic, epigenetic, hormonal, and environmental factors contribute to differences in outcomes to RSV infection between males and females. In addition to these, age-independent immunological differences can also play an important role. One key immunological factor linked to sex differences in viral infections is the differential expression of toll-like receptor 7 (TLR7) between males and females. TLR7 is a pattern recognition receptor that senses ssRNA in the endosomes and is therefore important for the detection of ssRNA viruses such as influenza (IAV), RSV, and SARS-CoV-2 [27]. Upon activation, it drives the production of proinflammatory cytokines and interferons (IFNs) from various innate immune cells and stimulates B cell proliferation, differentiation, and antibody production. As such, it is involved in stimulating innate and adaptive immune responses to infection. The *TLR7* gene is also located on the X chromosome and can escape X-inactivation, which results in female cells having heightened TLR7 expression [28]. Consequently, the production of type I IFNs and some proinflammatory cytokines is greater in female PBMCs and plasmacytoid dendritic cells upon TLR7 stimulation compared to male-derived cells [29,30]. This TLR7-mediated increase in the antiviral response might therefore be the reason why females display higher levels of resistance to SARS-CoV-2 infection [31]. Males tend to develop more severe COVID-19 disease than females, and reduced TLR7 expression in males has been implicated in this disparity [32]. Rare deleterious TLR7 variants, which result in poor type I IFN responses, have also been exclusively identified in males suffering from severe SARS-CoV-2 infection [33,34]. Furthermore, the expression of TLR7 in B cells is also higher in females and leads to higher antibody responses to IAV vaccination, which may explain the increased efficacy of vaccination in females compared to males [35].

The impact of sex differences on TLR7 responses to RSV infection is yet to be reported. Given that RSV shares a ssRNA genome similar to that of SARS-CoV-2 and IAV, it is plausible that sex differences in TLR7 expression could also influence the immunological responses to RSV infection. Previous work revealed that TLR7 enhanced disease severity following RSV infection in adult mice by promoting airway inflammation, lung pathology, and chronic airway hyperreactivity [36]. In a separate RSV model, deletion of TLR7 was shown to enhance goblet hyperplasia and mucous production [37]. While these studies did not specifically explore sex differences, they highlighted TLR7’s role in modulating the immune responses to RSV and influencing disease severity. Furthermore, experimental RSV mouse models reporting sex differences have exclusively focused on early-life infection [20,38,39]. To address this knowledge gap, we utilized an adult model of acute RSV infection to characterize sex differences in relation to infection and the effect of TLR7 deficiency on these responses, as TLR7 deficiency can minimize some of the age-related immunological differences that exist between males and females. A day 7 endpoint was selected for analysis, as it coincided with the onset of viral clearance and reflected the peak RSV load in the lung and nose, which typically occurs 4–6 days post infection [40]. This timepoint also corresponds with the emergence of early adaptive T cell and humoral responses, while marking the decline of innate immunity [36,40,41], providing a comprehensive overview of immune activity.

In response to acute RSV infection, we found that cytokine and antibody responses were heightened in the lungs of female mice compared to those of males. We also confirmed that TLR7 expression in the lungs was higher in females and that the deletion of TLR7 diminished sex differences in key immune responses. These results suggest that a better understanding of TLR7’s role could lead to more effective, sex-specific treatments for RSV.

## 2. Materials and Methods

### 2.1. Mice and Infections

Male and female wild-type (WT) C57BL/6J mice were obtained from the Animal Resources Centre (Perth, WA, Australia). Homozygous TLR7 knockout mice (B6.129S1-Tlr7tm1Flv/J) were obtained from The Jackson Laboratory (Bar Harbor, ME, USA) and bred in-house in the RMIT University animal research facility (Bundoora, VIC, Australia). Mice were housed in a 12 h light/12 h dark cycle, with ad libitum access to water and standard rodent chow (4.8% fat and 0.02% cholesterol). All animal experiments were conducted according to approval obtained from the RMIT University Animal Ethics Committee (Ethics number 23328) and in compliance with the guidelines of the National Health and Medical Research Council of Australia on animal experimentation.

For infection, 12–14-week-old male and female WT or TLR7ko mice were anesthetized by isoflurane inhalation and inoculated intranasally with 5 × 10^6^ plaque-forming units (PFU) of RSV-A Long strain or PBS (controls) in a 35 µL volume. The mice were weighed and monitored daily. After 7 days, the mice were euthanized by injection (i.p.) of a mixture of ketamine (180 mg/kg) and xylazine (32 mg/kg). RSV stocks were propagated and titrated on HEp-2 cells as previously described [36], with the virus resuspended in PBS + 1% fetal bovine serum (FBS; Sigma, Bayswater, VIC, Australia). The initial virus stock was kindly provided by Prof. Patrick Reading (Department of Immunology and Microbiology, The Peter Doherty Institute for Infection and Immunity, University of Melbourne).

### 2.2. Lung Gene Expression Analysis by qPCR

Total RNA was extracted from lung homogenates using the RNeasy Mini kit (Qiagen, Germantown, MD, USA). cDNA synthesis was performed on 2 μg of total RNA using the High-Capacity cDNA Reverse Transcription Kit (Applied Biosystems, Foster City, CA, USA) according to the following settings: 25 °C for 10 min, 37 °C for 120 min, 85 °C for 5 min. Quantitative polymerase chain reaction was carried out using the TaqMan Fast Advanced Master Mix (Thermofisher Scientific, Scoresby, VIC, Australia) and analyzed on a QuantStudio 7 Flex Real-Time PCR system (Thermofisher Scientific). The PCR primers used in this study were included in the Assay-on-Demand Gene Expression Assay Mix (Thermofisher Scientific), as well as the custom-designed forward and reverse oligonucleotides for the RSV Fusion gene 5′-TTGGATCTGCAATCGCCA-3′ and 5′-CTTTTGATCTTGTTCACTTCTCCTTCT-3′, using the Fast SYBR Green PCR Master Mix (Thermofisher Scientific). The following program settings were used for amplification: 50 °C for 2 min, 95 °C for 2 min, then 40 cycles at 95 °C for 1 s, and 60 °C for 20 s. Quantitative values were obtained from the average threshold cycle (Ct) number of each sample run in triplicate, and gene expression analysis was performed using the comparative Ct method. Target gene expression was normalized against RPS18 mRNA expression for each sample and expressed relative to the indicated control.

### 2.3. Assessment of Airway Inflammation and Differential Cell Counting

Bronchoalveolar lavage fluid (BALF) was obtained by lavaging the lungs to enumerate cells from the large and smaller airways and to assess inflammation. This involved creating an incision of the lower jaw to the top of the rib cage to expose the trachea. Another incision was made in the middle of the trachea, and a sheathed 21-gauge needle was inserted into the lumen. The lung was lavaged with 300–400 µL aliquots of PBS repeatedly, and the chest was gently massaged to further elute the aspirate until a collected volume of 1 mL was gained. The total number of live cells in the BALF was determined by acridine orange staining (Thermofisher Scientific), using a hemocytometer. Differential staining of cytospotted slides was also performed by Kwik-diff (Thermofisher Scientific), and the cell number was determined by counting 500 cells per slide based on standard microscopy morphological criteria.

### 2.4. Ex Vivo Infection of Alveolar Macrophages

Primary alveolar macrophages were obtained from the BALF of uninfected mice as described above. One hundred thousand cells were seeded per well in a 96-well plate in Dulbecco’s Modified Eagle Medium supplemented with L-glutamine, glucose (4.5 g/L), sodium pyruvate (110 mg/L), FBS (10%), and penicillin–streptomycin (1%) and allowed to adhere for 3–4 h. The cells were then infected with RSV (multiplicity of 10) or left untreated. After 24 h, the medium was aspirated, and the cells directly lysed for RNA extraction and subsequent gene expression analysis.

### 2.5. Measurement of Antibody Levels in BALF

BALF was extracted as above, the cells discarded after centrifugation at 400 g for 5 min, and the clear fluid snap-frozen and stored at −80 °C until use. The measurement of antibody isotypes IgA, IgE, IgG1, IgG2a, IgG2b, IgG3, and IgM was performed using the ProcartaPlex Mouse Antibody Isotyping Panel 7-Plex Kit (Thermofisher Scientific, cat. no. EPX070-20815-901). Briefly, 50 µL of undiluted BALF was analyzed in duplicate, and incubations were performed according to the manufacturer’s instructions. The plates were read on a Bio-Plex 200 instrument (Bio-Rad, Gladesville, NSW, Australia), with 50 counts per region, and the median fluorescence intensity was measured.

### 2.6. Statistical Analysis

All data are expressed as the mean ± SEM. All comparisons were performed using GraphPad Prism (GraphPad Software Version 10.1.2, Boston, MA, USA) to calculate two-way ANOVA followed by post-hoc tests for multiple comparison (specified in the figure legends). Statistical significance was considered at *p* < 0.05.

## 3. Results

### 3.1. Female Mice Exhibit Higher Inflammatory Gene Expression in the Lungs Following Acute RSV Infection

Our first aim was to identify if there were sex differences in lung inflammatory markers in response to RSV infection in adult wild-type (WT) mice. Male and female mice were infected with RSV, and body weight was recorded until 7 days post infection. Acute infection did not induce body weight loss in either sex (Appendix A).

We then compared the gene expression of various inflammatory markers in the lungs at the 7-day endpoint, comparing infected and uninfected mice and males and females. Following infection, female mice exhibited significant increases in the proinflammatory genes *IL1B*, *IL6*, and *TNFA* (Figure 1A). The upregulation of *IL1B* upon infection was absent in male mice, while the *IL6* levels were higher relative to those in uninfected male mice, although significantly lower compared to those in the infected female group. *TNFA* expression was also elevated with infection in males and significantly higher than in female mice. An increase in the expression of Th2 cytokines was also identified as follows: increases in *IL4* and *IL13* but not in *IL5* in female mice and increases in *IL5* and *IL13* but not in *IL4* in male mice (Figure 1B). Male mice showed an overall lower *IL4* and *IL13* response compared to female mice. The upregulation of *IL5* was only observed in male mice. We also measured type I, II, and III IFN as well as *IRF7* expression in the lungs. The expression of all IFNs increased with infection in both sexes (Figure 1C). The *IFNB*, *IFNG*, and *IRF7* levels were significantly lower in infected male mice compared to female mice, while no difference in *IFNL3* expression was observed between male and female mice. Infection also boosted the expression of the *CXCL2*, *CCL2*, *CCL3*, and *CCL5* chemokines in female mice, while only *CCL2* and *CCL3* expression increased in males (Figure 1D). The upregulation of these chemotactic markers in male mice was significantly lower compared to that observed in female mice. RSV transcripts were also detected in the lungs of both sexes, although they were significantly less abundant in male compared to female mice (Figure 1E).

This analysis revealed that female mice display significantly higher proinflammatory, Th2, IFN, chemokine, and viral transcript levels in response to RSV infection when compared to males.

### 3.2. TLR7 Deficiency Reduces Cytokine Gene Expression in Response to Acute RSV Infection in Both Sexes

A possible reason for the observed sex differences in the inflammatory cytokine response could be variations in the basal gene expression of TLR7. Our analysis of gene expression in the lungs of uninfected mice showed that females had a basal expression of TLR7 that was significantly 10-fold higher than in males (Figure 2). Expression was undetectable in TLR7 knockout (TLR7ko) mice.

We next measured the gene expression profile in the lungs of male and female TLR7ko mice following infection with RSV. As with WT mice, infection of TLR7ko mice did not provoke significant changes in body weight in either sex (Appendix A). In female TLR7ko mice, the expression of inflammatory cytokines (Figure 3A), except for *TNFA*, and Th2 cytokines (Figure 3B) was unaltered by infection, while the levels of type I/III IFNs (Figure 3C) and some chemokines (Figure 3D) were significantly elevated relative to those in the uninfected female group. Of the gene markers analyzed, male TLR7ko mice only showed significant increases in *IL5* and *IRF7* expression following infection. Similar to WT mice, significantly more viral transcripts were detected in the lungs of female compared to male TLR7ko mice (Figure 3E).

A net reduction in most Th1, Th2, and IFN gene responses to RSV infection was identified in TLR7ko mice of both sexes compared to WT mice, indicating a TLR7-dependent effect for those genes (Table 1). We noted that infection in female TLR7ko mice boosted *IFNG*, *CCL2*, *CCL3*, and *CCL5* expression, although significantly less compared to the female WT group, but these responses were completely blunted in TLR7ko male mice. This suggests that the upregulation of these particular genes is solely TLR7-dependent in male mice. Furthermore, the *IL5* response to infection in male mice was not TLR7-dependent. Interestingly, TLR7 deletion eliminated the sex differences specifically in the *IL1B*, *IL6*, *IL4*, *IL13*, *IFNB*, *IRF7*, and *CXCL2* responses.

This analysis indicated that TLR7 is more highly expressed in female lungs and is required for RSV-induced cytokine responses, and its deletion abolished the sex differences in some of these responses.

### 3.3. Female Mice Exhibit a Greater Influx of Macrophages into the Airways Following Acute RSV Infection

We next investigated the immune cell types infiltrating the airways that might be involved in the sex differences in the cytokine responses described above. Similar numbers of live cells were detected in the bronchoalveolar lavage fluid (BALF) of female and male mice following RSV infection, indicating a similar degree of airway inflammation (Figure 4A). To determine whether there were any alterations in specific cell types, the cells were differentially stained, and four cell types were identified by standard morphological criteria. Following infection, WT mice of both sexes showed increases in macrophages, neutrophils, lymphocytes, and eosinophils. Macrophage counts in WT females were significantly higher than in WT males, while no significant sex differences were found in the other cell types. TLR7ko mice showed a reduction in the number of all immune cell types, regardless of sex, and no sex difference in macrophage infiltration (Figure 4B). Furthermore, deletion of TLR7 introduced a sex difference in the response of other immune cells, with lower neutrophil counts and higher lymphocyte counts in males. This analysis revealed that RSV infection in female mice promoted increased macrophage infiltration in the airways compared to males and that this effect was TLR7-dependent.

### 3.4. RSV Infection of Male Alveolar Macrophages Induces a Stronger Proinflammatory Cytokine Response than in Female Cells

The findings above indicate a sex difference in macrophage infiltration following RSV infection. While this could be due to the increased macrophages/monocyte chemotactic expression of *CCL2* and *CCL3* in female lungs (Figure 1D), it may also reflect differences in how macrophages themselves directly respond to the virus. To investigate potential sex differences in macrophages, we ex vivo infected primary alveolar macrophages from naïve male and female mice with RSV and measured their cytokine responses (Figure 5). Following infection, macrophages of both sexes showed similar increases in *TLR7* and *IFNB* expression, while significant upregulation of *IL6* and *TNFA* was evident only in male macrophages. Despite using the same viral inoculum, the male macrophages contained significantly more RSV transcripts than their female counterparts. This revealed intrinsic sex differences in the proinflammatory response but not in type I IFN expression by macrophages following direct RSV infection.

### 3.5. Female Mice Exhibit Higher Antibody Titers Following Acute RSV Infection

Finally, we measured antibody responses in the BALF following acute RSV infection and compared male and female mice. RSV infection led to increased antibody titers for all isotypes tested in both male and female WT mice, except for IgE and IgG3, whose levels were only increased in females (Figure 6A). Significantly less abundant IgG2a, IgG2b, and IgM were found in male mice compared to females. TLR7 deficiency in female mice significantly blunted or reduced the titers of all antibody isotypes (Figure 6B). In contrast, male TLR7ko mice showed a diminished IgA, IgG2a, and IgM response, while the IgE, IgG1, IgG2b, and IgG3 levels remained unchanged or were higher in males. Deletion of TLR7 eliminated the sex differences associated with the IgG2a, IgG2b, and IgM isotypes but introduced a dominant IgE and IgG3 response in males.

This analysis revealed sex differences in antibody responses in the airways of RSV-infected mice. TLR7 deficiency reduced the antibody titers mostly in female mice and eliminated some of the sex differences.

## 4. Discussion

In this study, we gained new insights into the mechanisms driving sex differences in RSV pathophysiology by showing a role for TLR7 in influencing the immune responses to infection by this virus. Our results demonstrated that females exhibited a more robust inflammatory response to RSV infection, marked by higher expression of various inflammatory cytokines and IFNs and higher antibody titers in the lungs compared to males. Importantly, we showed that TLR7 plays a significant role in driving these sex differences, with females having higher basal expression of TLR7 in the lungs, and the deletion of TLR7 eliminating many of the observed immune response disparities (Appendix A). Furthermore, our data indicate that TLR7-mediated effects are also seen in immune cell infiltration, particularly for macrophages, which were more abundant in the airways of female mice following RSV infection.

Our findings align with previous studies suggesting that females mount stronger immune responses to respiratory infections, which often leads to less severe outcomes compared to those observed in males [35,42,43]. However, our results showed that this heightened immune response is highly TLR7-dependent. This enhanced immune response in females may be beneficial in accelerating viral clearance and stimulating more robust adaptive immune responses. While this could render males more vulnerable to severe effects of the infection, it may also contribute to reduced immunopathology and a lower risk of autoimmunity. For example, females appear to have a higher susceptibility to autoimmune diseases [44] such as systemic lupus erythematosus, partly due to the higher expression and activation of TLR7 in B cells, which predisposes them to autoimmune conditions [45]. Furthermore, increased autoimmunity has been observed in female patients with other respiratory infections, such as COVID-19, where higher levels of autoantibodies are found in females compared to males. [46,47]. Exacerbated TLR7 activity has also been implicated in chronic respiratory conditions including chronic airway hyperreactivity following RSV infection [36] and COPD-like pathology [48]. Given that the majority of non-smoker COPD patients are female [49], a potential TLR7-dependent sex bias may also exist in COPD. Overall, while TLR7-mediated enhanced antiviral immunity in females can be beneficial, the chronic pathological effects of persistent TLR7 activation on immune tolerance and immune-mediated diseases must be considered when examining TLR7-dependent sex differences in disease.

To our knowledge, this is the first study to report the effect of TLR7 on immune responses to acute RSV infection and whether this underpins sex-related differences. Deletion of TLR7 reduced the overall lung inflammatory response to infection, airway inflammation, and antibody response in both male and female mice. This is consistent with TLR7-mediated hyperinflammation worsening disease outcomes following infection with other ssRNA viruses [50,51]. Notably, we found that TLR7 deficiency eliminated the female bias in gene responses, specifically for *IL1B, IL6, IL4, IL13, IFNB, IRF7*, and *CXCL2*, as well as in macrophage airway infiltration and IgG2a, IgG2b, and IgM titers in response to RSV infection. This suggests that the higher viral load in female lungs, which was not significantly affected by TLR7 deletion, is not solely responsible for the stronger immune response in females. Given the increased basal expression of TLR7 in female lungs, our findings indicate that TLR7 can amplify proinflammatory cytokine, IFN, and antibody responses to RSV infection in females.

Among the cell types that we considered in our analysis, more macrophages were recruited to the airways of female mice following infection. This may reflect increased macrophage chemotaxis in female lungs in response to RSV infection. However, it is also possible that macrophages themselves, which express substantial amounts of TLR7, exhibit altered responses to direct infection depending on their sex of origin. Indeed, the expression of TLRs is higher in female macrophages [52], and dysregulated TLR7 signaling in macrophages can drive female-biased disease [53]. We found that direct RSV infection of male alveolar macrophages produced more proinflammatory cytokines, while the type I IFN levels were comparable to those in female cells. These findings contrast with the enhanced type I IFN transcriptional profile in female bone marrow-derived dendritic cells following RSV infection compared to male cells [38]. However, this may simply reflect the different immune roles played by these cell types in response to infection. Interestingly, similar male-biased proinflammatory responses were observed in monocytes infected with SARS-CoV-2 [54], and male macrophages or monocytes also showed higher production of IL1B and TNF cytokines when stimulated with LPS, compared to female-derived cells [55,56,57]. These findings are in contrast to the female bias in cytokine responses in our in vivo lung analysis, although *TNFA* was one of the only genes in the lungs whose expression was significantly higher in males compared to females. It is possible, however, for the TLR7-mediated sex difference not to exist solely at the inflammatory cell level, as TLR7 expression can also be higher in some female non-immune cells [58]. Clinical evidence implicates testosterone in potentiating proinflammatory responses, including TNF, and attenuating type I IFN responses [54], highlighting the role of sex hormones in immune system adaptations in vivo, which are probably absent in most ex vivo models. Future studies exploring how hormonal fluctuations could influence the immune response in multiple cell types, specifically in the context of RSV infection, would offer further insight into these differences [59].

Such factors may also explain the sex differences we identified that were not mediated by TLR7 such as the heightened *IFNG, CCL2, CCL3, CCL5*, and viral transcripts in female TLR7ko mice. The X chromosome contains a high density of immune-related genes and regulatory elements [60]; so, the differential expression of other X-linked immune-related genes may also contribute to altered immune responses in the two sexes. For instance, female T cells can express more *CD40LG* or *CXCR3*, both of which are X-linked, and are generally more responsive to activation, produce more effector molecules like IFNγ in response to infection, and generate more short-lived effector cells than male-derived T cells [55,61,62]. TLR7 was not required for T cell effector function following IAV infection [63], although estrogen, which is found in higher concentrations in females, has been shown to directly boost the expression of T cell effector genes in female cells [64]. The RSV-induced *IL5* response, observed only in male mice, was also independent of the TLR7 genotype. Changes in testosterone levels can alter the Th2/IL17A responses in the lung and affect airway hyperreactivity [65,66], which may explain this observation.

We also considered the possibility that sex-dependent humoral responses to RSV infection might be governed by TLR7. Enhanced TLR7 expression was responsible for the improved antibody responses to IAV infection and vaccination in female mice compared to males [35]. Consistent with this, we found an overall stronger antibody response in female mice exposed to RSV, except for IgA. Although we did not investigate long-term antibody or memory responses, our study suggests that the heightened humoral immunity during acute RSV infection in females could offer additional protection against the current infection. Interestingly, while TLR7 was important for the antibody response in both males and females, an opposing male bias was observed upon TLR7 deletion, specifically for the IgE and IgG3 isotypes. This suggests that males on a TLR7-deficient background may be more susceptible to impairments in isotype switching following RSV infection, which may lead to increased production of IgE and IgG3. Indeed, TLR7 deficiency altered IgG class switching following IAV infection [63], and this was associated with an increased production of Th2 cytokines [51,67]. High IgG3 class types are associated with severe COVID-19 [68], and enhanced IgE production is involved in Th2-driven hyperresponsive allergic disease, eosinophilia, and mucus hyperproduction [69]. The activation of TLR7 boosts Th1 immunity to counteract a Th2 phenotype by suppressing IgE, IL4, IL5, and IL13 cytokine production [70,71,72]. As such, defects in TLR7 expression or function are overrepresented in individuals with asthma, resulting in enhanced IgE and Th2 responses underlying the disease [73,74,75,76]. Furthermore, the specific susceptibility of male mice to allergic disease following early-life RSV infection was associated with a heightened and delayed Th2/Th17 immune response [20]. Alterations in the gut microbiome composition were also most pronounced in male mice following RSV infection, which has implications for T cell immune tolerance and allergies [39]. Our study therefore suggests that males, which express less TLR7, may rely more heavily on the Th2-suppressive effects of TLR7 during RSV infection for protection against chronic airway disease.

It is important to acknowledge that the levels of IgE and the Th2 cytokines *IL4* and *IL13* were also elevated in RSV-infected females, suggesting that these mice could be more vulnerable to allergic asthma. However, this Th2 response in females was accompanied by a parallel elevation in the levels of Th1 cytokines and other antibody isotypes, implying that this was a general feature of the total cytokine response. Subsequent studies evaluating the functional consequence of this would be valuable to address the chronic consequences of these different Th2 profiles in female and male mice. Interestingly, opposite to males, the IgE and Th2 responses were suppressed in female TLR7ko mice. Based on this, restraining TLR7 activity may benefit females more by providing protection against chronic airway disease following RSV infection [36].

Overall, our findings suggest that the role of TLR7 in RSV disease may differ between males and females. While TLR7 drives antiviral and inflammatory responses to infection in both sexes, females tend to have stronger inflammatory and antibody responses. This may improve the management of the infection but might also increase the susceptibility for immune-mediated pathology and autoimmunity. On the other hand, in males, the intensified Th1 response could suppress Th2-driven disease, potentially increasing their risk of airway hyperresponsiveness if TLR7 activity is reduced. This difference in TLR7 activity could be a reason why males are more susceptible to asthma and wheezing following RSV infection [77]. Although yet to be explored in this context, it is possible for other immune pathways to also contribute to the observed sex differences with respect to RSV infection. For instance, TLR8, which is also located on the X chromosome in close proximity to TLR7 and similarly senses ssRNA, can also evade X-inactivation, which results in its more abundant expression in female cells and differential cytokine responses compared to males [78,79]. Further, expression of RIGI, another sensor of viral RNA when present in the cytosol, was also higher in female T cells [64].

Our study underscores the importance of considering sex differences as a biological variable in RSV research and highlights the potential for targeting TLR7 pathways for therapeutic strategies. While age does not affect the stimulatory capacity of TLR7, females show stronger cytokine production due to higher cellular expression of TLR7 [80]. This suggests that the effectiveness of synthetic TLR7 ligands could be greater in females than in males, especially in combating initial RSV infection. Future studies should therefore explore how TLR7 modulation could be leveraged to develop temporal targeted therapeutic interventions for both sexes.

## Figures and Tables

**Figure 1 viruses-17-00428-f001:**
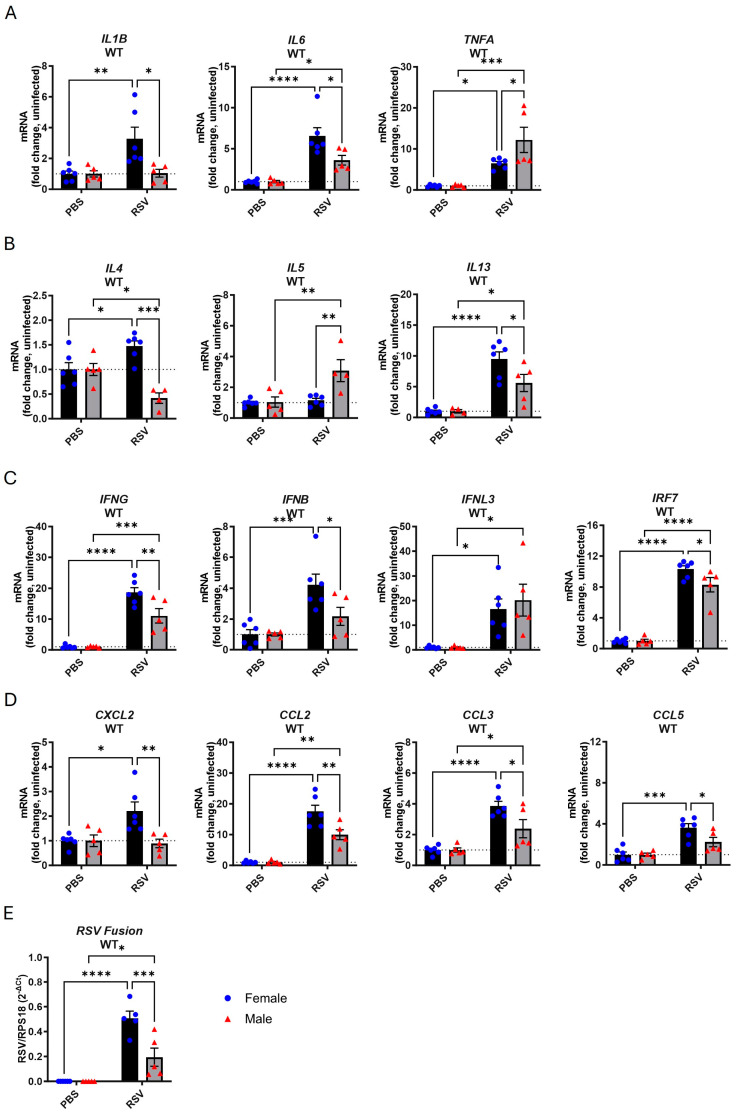
Increased inflammatory gene expression in the lungs of female WT mice following RSV infection. Male or female WT C57Bl/6 mice were infected with RSV-A Long (5 × 10^6^ PFUs) or mock-infected with PBS. Lung tissue was isolated after 7 days, and gene expression of (**A**) proinflammatory cytokines, (**B**) Th2 cytokines, (**C**) IFNs, and (**D**) chemokines analyzed by RT-qPCR. Gene expression is presented relative to RPS18 housekeeping gene expression as a fold change with respect to PBS controls. (**E**) Amplification of RSV Fusion gene is presented relative to that of RPS18 housekeeping gene (2^−ΔCt^). Data are expressed as mean ± SEM, *n* = 5–6. Statistical analysis was conducted using two-way ANOVA test followed by Tukey’s post hoc test for multiple comparisons (* *p* < 0.05, ** *p* < 0.01, *** *p* < 0.001, **** *p* < 0.0001).

**Figure 2 viruses-17-00428-f002:**
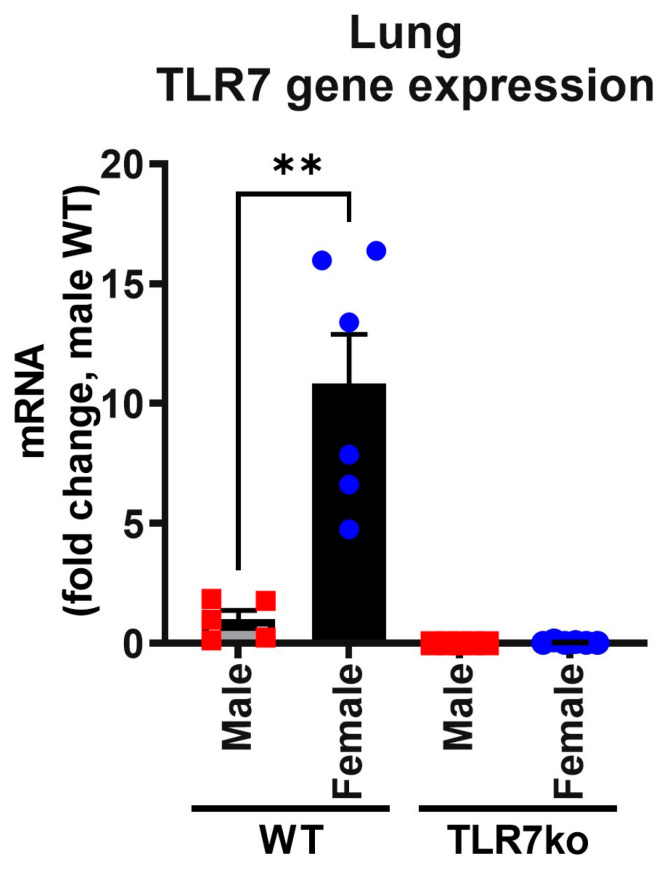
TLR7 gene expression is higher in female lungs. Gene expression of TLR7 in the lungs of male or female WT C57Bl/6 and TLR7ko mice was measured by RT-qPCR. Data are presented relative to RPS18 housekeeping gene expression as a fold change with respect to the WT male group. Data are expressed as mean ± SEM, *n* = 5–6. Statistical analysis was conducted using unpaired Student’s *t*-test (** *p* < 0.01).

**Figure 3 viruses-17-00428-f003:**
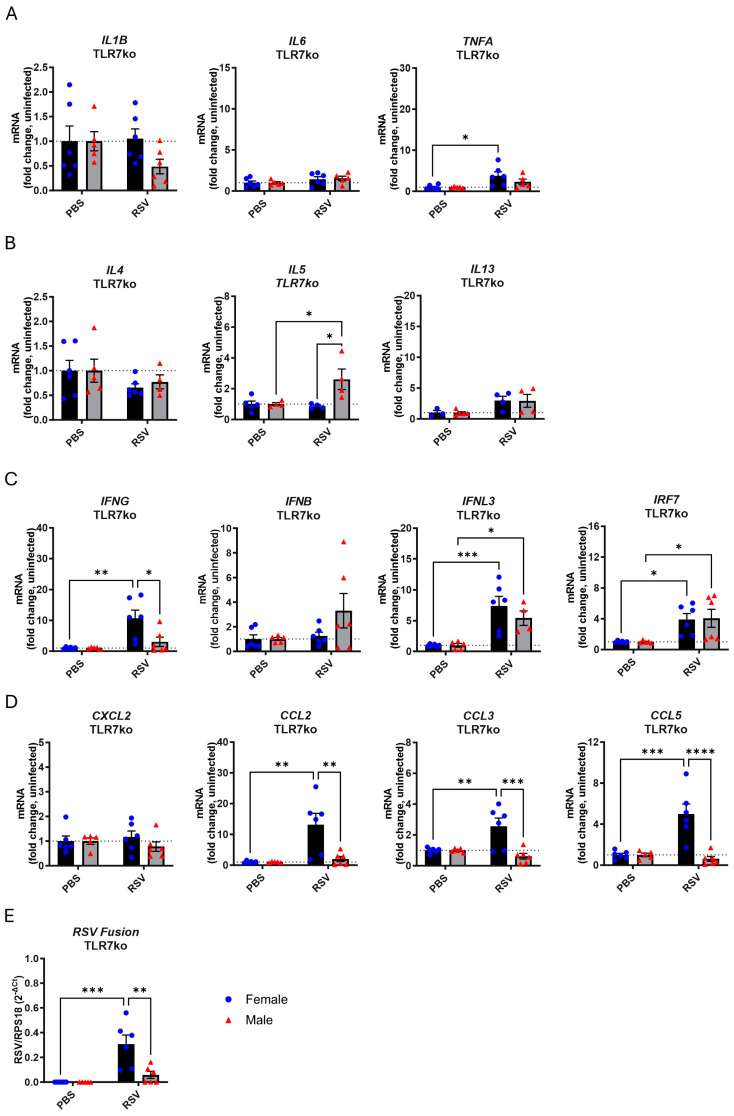
TLR7ko mice exhibit reduced inflammatory expression in the lungs following RSV infection. Male or female TLR7ko mice were infected with RSV-A Long (5 × 10^6^ PFUs) or mock-infected with PBS. Lung tissue was isolated after 7 days, and gene expression of (**A**) proinflammatory cytokines, (**B**) Th2 cytokines, (**C**) IFNs, and (**D**) chemokines analyzed by RT-qPCR. Gene expression is presented relative to RPS18 housekeeping gene expression as a fold change with respect to PBS controls. (**E**) Amplification of RSV Fusion gene is presented relative to RPS18 housekeeping gene expression (2^−ΔCt^). Data are expressed as mean ± SEM, *n* = 4–6. Statistical analysis was conducted using two-way ANOVA test followed by Tukey’s post hoc test for multiple comparisons (* *p* < 0.05, ** *p* < 0.01, *** *p* < 0.001, **** *p* < 0.0001).

**Figure 4 viruses-17-00428-f004:**
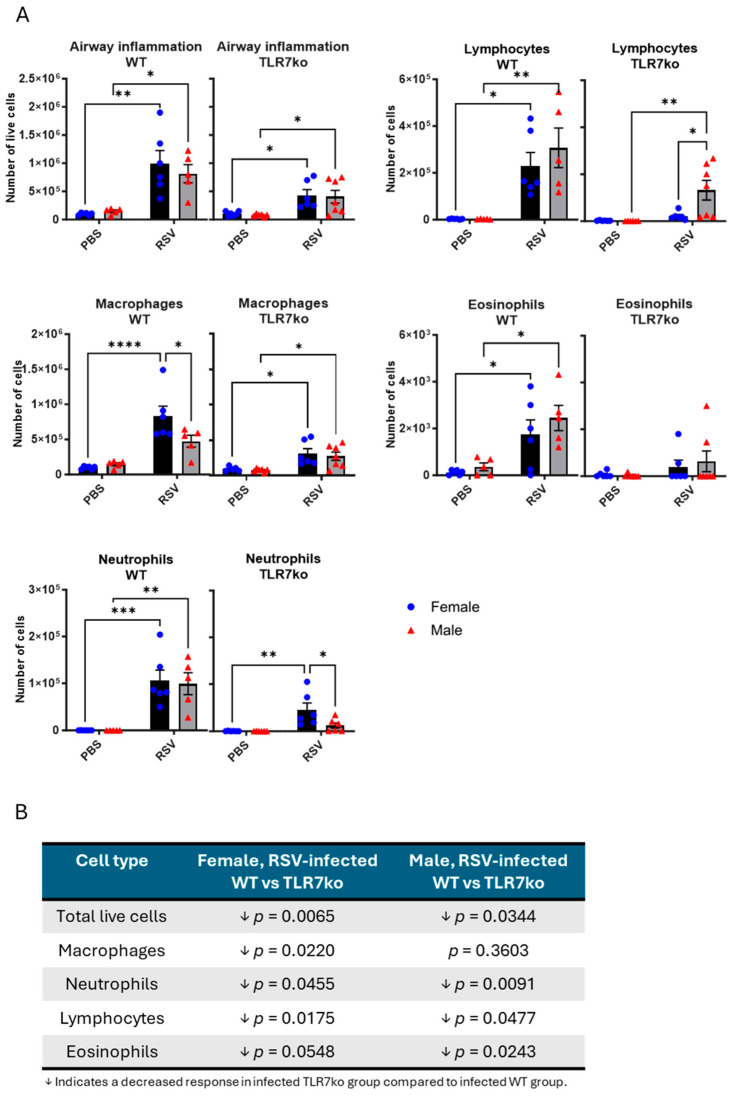
Female WT mice exhibit increased macrophage infiltration in the airways following RSV infection. Male or female TLR7ko mice were infected with RSV-A Long (5 × 10^6^ PFUs) or mock-infected with PBS. (**A**) BALF was obtained after 7 days, and the number of live cells counted. The numbers of differentially stained macrophages, neutrophils, lymphocytes, and eosinophils was determined by counting 500 cells in random fields by standard morphological criteria relative to the total number of isolated cells. (**B**) Statistical comparison of the number of cell types between WT and TLR7ko mice is shown in the table. Data are expressed as mean ± SEM, *n* = 5–6. Statistical analysis was conducted using two-way ANOVA test followed by Tukey’s post hoc test for multiple comparisons (* *p* < 0.05, ** *p* < 0.01, *** *p* < 0.001, **** *p* < 0.0001).

**Figure 5 viruses-17-00428-f005:**
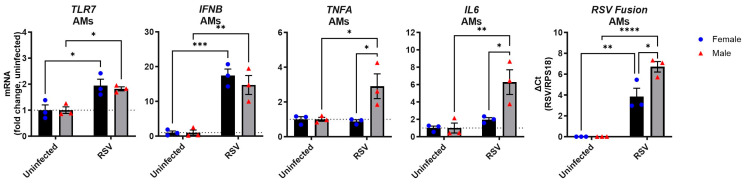
Male alveolar macrophages exhibit increased proinflammatory cytokine expression following RSV infection. Primary alveolar macrophages (AMs) were isolated from male and female naïve WT mice and infected ex vivo with RSV (MOI 10) for 24 h. Expression of indicated genes was then performed and is presented relative to RPS18 expression as a fold change with respect to uninfected controls. Data are expressed as mean ± SEM, *n* = 3. Statistical analysis was conducted using two-way ANOVA test followed by Tukey’s post hoc test for multiple comparisons (* *p* < 0.05, ** *p* < 0.01, *** *p* < 0.001, **** *p* < 0.0001).

**Figure 6 viruses-17-00428-f006:**
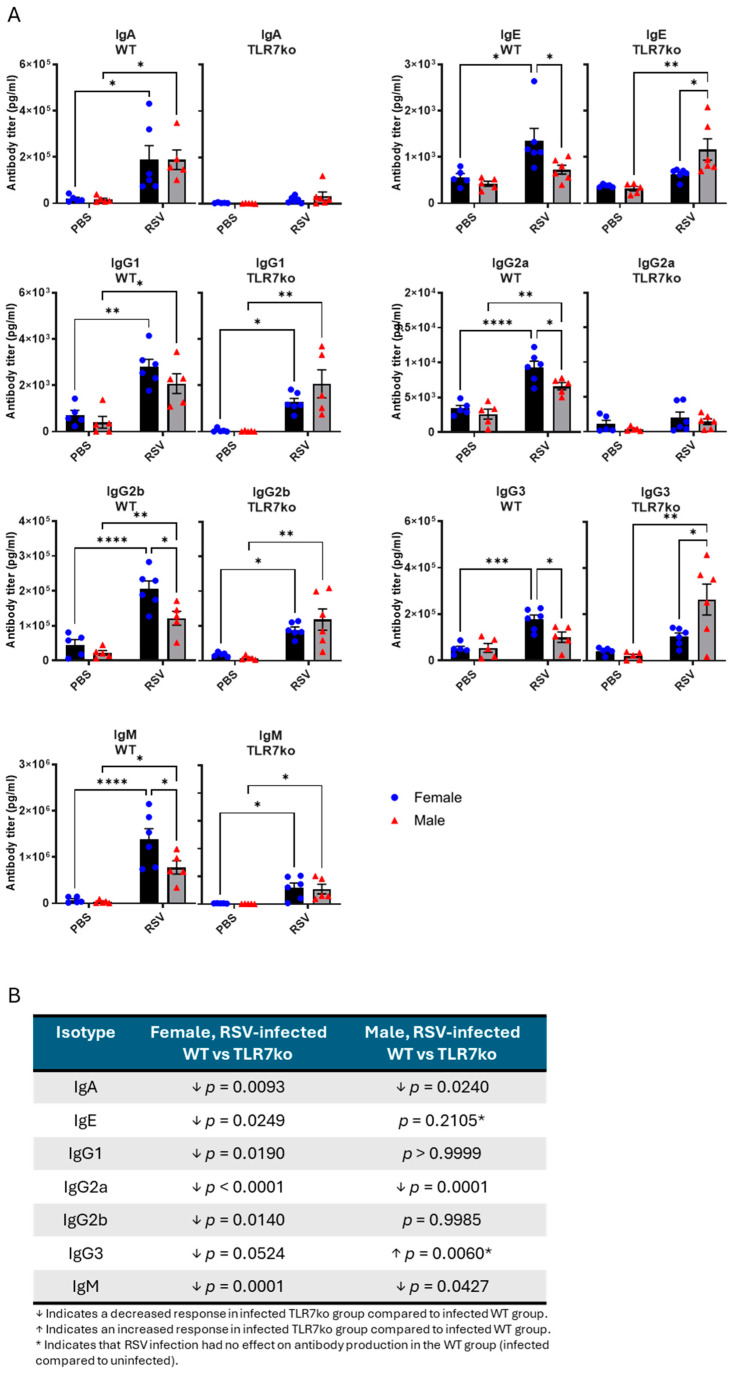
Female WT mice exhibit increased antibody titers in the airways following RSV infection. Male or female WT C57Bl/6 or TLR7ko mice were infected with RSV-A Long (5 × 10^6^ PFUs) or mock-infected with PBS. (**A**) Antibody isotyping in the BALF after 7 days was determined by multiplex assays. (**B**) Statistical comparison of the antibody titer between WT and TLR7ko mice is shown in the table. Data are expressed as mean ± SEM, *n* = 5–6. Statistical analysis was conducted using two-way ANOVA test followed by Tukey’s post hoc test for multiple comparisons (* *p* < 0.05, ** *p* < 0.01, *** *p* < 0.001, **** *p* < 0.0001).

**Table 1 viruses-17-00428-t001:** Statistical comparison of lung gene expression between infected WT and TLR7ko mice.

Gene	Female, RSV-Infected WT vs. TLR7ko	Male, RSV-Infected WT vs. TLR7ko
IL1B	↓ *p* = 0.0025	*p* = 0.6172 *
NLRP3	↓ *p* < 0.0001	↓ *p* = 0.0431 *
IL18	↓ *p* = 0.0938	*p* = 0.8545 *
IL6	↓ *p* = 0.0004	*p* = 0.1204
TNFA	↓ *p* = 0.0731	↓ *p* = 0.0020
IL4	↓ *p* < 0.0001	↓ *p* = 0.0961 *
IL5	*p* = 0.8696 *	*p* = 0.7319
IL13	↓ *p* = 0.0028	*p* = 0.2654
IFNG	↓ *p* = 0.0144	↓ *p* = 0.0217
IFNB	↓ *p* = 0.0466	*p* = 0.6202 *
IFNL3	*p* = 0.1950	↓ *p* = 0.0554
IRF7	↓ *p* < 0.0001	↓ *p* = 0.0066
CXCL2	↓ *p* = 0.0200	*p* = 0.9447 *
CCL2	*p* = 0.3583	↓ *p* = 0.0612
CCL3	↓ *p* = 0.0154	↓ *p* = 0.0123
CCL5	*p* = 0.2243	*p* = 0.1553 *
RSV F	↓ *p* = 0.0866	*p* = 0.1996

↓ Indicates a decreased response in the infected TLR7ko group compared to the infected WT group. * Indicates that RSV infection had no effect on gene expression in the WT group (infected compared to uninfected).

## Data Availability

The data presented in this study are available in this article and the relevant Appendix A.

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
