# Peer review of "Exploring the Contribution of TLR7 to Sex-Based Disparities in Respiratory Syncytial Virus (RSV)-Induced Inflammation and Immunity"

_viruses, 2025, doi:10.3390/v17030428_

Round 1

Reviewer 1 Report

Comments and Suggestions for Authors

The authors present a comprehensive study investigating the sex-dependent role of TLR7 in specific lung immune responses upon acute RSV infection, using adult C57Bl/6 andr TLR7 knockout mice. The results provide evidence for an important role of TLR7 in enhancing inflammatory response in females. The study is well conducted resulting in reasonable conclusions.

As an amendment, (i) the food composition and (at least approximate food intake) of the mice and (ii) the type and source of the applied RSV-A Long strain should be indicated. Concerning the latter, the presence of additional components in the inoculated RSV sample should be indicated as well as whether it represents a a homogeneous suspension of virus particles confirmed by electron microscopy.

Author Response

Reviewer #1:

The authors present a comprehensive study investigating the sex-dependent role of TLR7 in specific lung immune responses upon acute RSV infection, using adult C57Bl/6 andr TLR7 knockout mice. The results provide evidence for an important role of TLR7 in enhancing inflammatory response in females. The study is well conducted resulting in reasonable conclusions.

As an amendment, (i) the food composition and (at least approximate food intake) of the mice and (ii) the type and source of the applied RSV-A Long strain should be indicated. Concerning the latter, the presence of additional components in the inoculated RSV sample should be indicated as well as whether it represents a a homogeneous suspension of virus particles confirmed by electron microscopy.

Author response:

We thank this reviewer for their time and assessment of our manuscript.

We have added the following additional information to the Methods section 2.1:

  • Lines 135-136: “Mice were housed in a 12 h light/12 h dark cycle with ad libitum access to water and standard rodent chow (4.8% fat and 0.02% cholesterol).”
  • Lines 144-148: “RSV stocks were propagated and titrated on HEp-2 cells as previously described [36], and virus resuspended in PBS + 1% fetal bovine serum (FBS; Sigma, Bayswater, VIC, Australia). The initial virus stock was kindly provided by Prof. Patrick Reading (Department of Immunology and Microbiology, The Peter Doherty Institute for Infection and Immunity, University of Melbourne).”

We had not performed electron microscopy to determine whether the RSV stock contained a homogeneous suspension of virus particles.

Reviewer 2 Report

Comments and Suggestions for Authors

Dear Authors,

I would like to congratulate you on preparing an interesting manuscript. However, after reviewing the manuscript, I have identified a few areas that need minor corrections or clarifications.

In the introduction, you mentioned that prevention strategies against RSV have been developed for older age groups. I think adding short information on the anti-RSV vaccine would benefit the manuscript. Also, a severe RSV infection in infants is preventable, either through maternal RSV vaccination or infant anti-RSV monoclonal antibody administration.

I would like to propose enhancing the quality of the figures. The text on the figures lacks sufficient sharpness. 

The manuscript refers to the Supplementary Materials associated with this submission. Regrettably, I did not receive supplementary files pertinent to the primary manuscript text.

Author Response

Reviewer #2:

I would like to congratulate you on preparing an interesting manuscript. However, after reviewing the manuscript, I have identified a few areas that need minor corrections or clarifications.

In the introduction, you mentioned that prevention strategies against RSV have been developed for older age groups. I think adding short information on the anti-RSV vaccine would benefit the manuscript. Also, a severe RSV infection in infants is preventable, either through maternal RSV vaccination or infant anti-RSV monoclonal antibody administration.

I would like to propose enhancing the quality of the figures. The text on the figures lacks sufficient sharpness.

The manuscript refers to the Supplementary Materials associated with this submission. Regrettably, I did not receive supplementary files pertinent to the primary manuscript text.

Author response:

We thank this reviewer for their time and assessment of our manuscript.

In response, we have added the following additional text to the Introduction section:

  • Lines 43-49: “The first RSV vaccines, Abrysvo, Arexvy and mResvia, were recently approved for use in older adults; showing a significant reduction in RSV-associated LRT disease in over 60 year olds [5-7]. For infants, vaccination during pregnancy offers some protection for new-borns during their early months of life [8]. Prophylactic use of RSV-neutralizing mono-clonal antibodies like Palivizumab and Nirsevimab can also provide protection in young infants, but these treatments are reserved for high-risk cases, and their effects diminish relatively quickly [9,10].”

We will also work with the journal’s production team to ensure the figures are of the highest quality in the final article.